# A mixed method study design to explore the adherence of haematological cancer patients to oral anticancer medication in a multilingual and multicultural outpatient setting: The MADESIO protocol

Sandra Michiels[1,2,3]*, Sandra Tricas-Sauras[2], Marie Dauvrin[4,5,6], Dominique Bron[3], Fati Kirakoya-Samadoulougou[1]

**1** Centre for Research in Epidemiology, Biostatistics, and Clinical Research, School of Public Health, Université Libre de Bruxelles, Brussels, Belgium, **2** Centre for Research in Social Approaches to Health, School of Public Health, Université Libre de Bruxelles, Brussels, Belgium, **3** Department of Haematology/ Oncology, Institut Jules Bordet, Université Libre de Bruxelles, Brussels, Belgium, **4** Department of Public Health, Université Catholique de Louvain, Brussels, Belgium, **5** Belgian Health Care Knowledge Centre, Brussels, Belgium, **6** Haute Ecole Léonard de Vinci, Brussels, Belgium

* sandra.michiels@ulb.be, sandra.michiels@bordet.be

**Funding:** The first author, SM, obtained two research grants from the Fondation Kisane and les

## Abstract

### Background

Patients with haematologic malignancies are increasingly treated by oral anticancer medications, heightening the challenge of ensuring optimal adherence to treatment. However, except for chronic myelogenous leukaemia or acute lymphoid leukaemia, the extent of non-adherence has rarely been investigated in outpatient settings, particularly for migrant population. With growing numbers of migrants in Belgium, identifying potential differences in drug use is essential. Also, previous research regarding social determinants of health highlight important disparities for migrant population. Difficulties in communication between health caregivers and patients from different cultural and ethnic backgrounds has been underlined.

### Methods

Using a sequential mixed method design, the MADESIO protocol explores the adherence to oral anticancer medications in patients with haematological malignancies and among first and second generation migrants of varied origin. Conducted in the ambulatory setting, a first quantitative strand will measure adherence rates and associated risk factors in two subgroups of patients with haematological malignancies (group A: first and second generation migrants and group B: non-migrants). The second qualitative strand of this study uses semi-structured interviews to address address the patients' subjective meanings and understand the statistical associations observed in the quantitative study (strand one). MADESIO aims to provide a first assessment of whether and why migrants constitute a population at risk concerning adherence to oral anticancer medications.

Amis de l'Institut to cover the part-time dedicated to the research. In 2018 the study project was awarded by the Belgian Society of Haematology as part of a broader project aiming to support migrant patients and won the PAtient CEntricity Award (PACE Award). The 10.000 € PACE Award allowed to cover the purchasing licensing fees for the use of the questionnaires and the translation fees. The funders had and will not have a role in study design, data collection and analysis, decision to publish, or preparation of the manuscript.

**Competing interests:** The authors have declared that no competing interests exist.

## Discussion

Our protocol is designed to provide a comprehensive understanding of adherence in a specific population. The methodological choices applied allow to explore adherence among patients from diverse linguistic and cultural backgrounds. A particular emphasis has been paid to minimize the biases and increase the reliability of the data collected. Easily reproductible, the MADESIO design may help healthcare services to screen adherence to Oral anticancer medications and to guide providers in choosing the best strategies to address medication adherence of migrants or minority diverse population.

## 1. Background

For the last two decades, the use of oral anticancer medications (OAMs) has been increasing, and oncological teams must deal with the challenge of ensuring optimal adherence to prescribed treatments. Poor adherence impacts the quality of response and the risk of relapse [1, 2]. However, data are scarce regarding how cancer patients adhere to their medication plan, particularly those with malignancies other than breast cancer [3].

Moreover, oncohaematology appears to be the poor cousin to cancer adherence studies. Of the 51 studies involved in the latest 2016 systematic review [4] focusing to adherence to OAMs, 30 focused on breast cancer compared to 9 on haematological cancer patients, all pertaining to patients with either chronic myelogeneous leukemia (CML) or acute lymphoïd leukemia (ALL).

Due to the wide variety of methods used, the lack of standardisation in defining optimal adherence and the considerable differences in length of study follow-up or methodological quality, adherence values range widely across studies. Although no reliable estimate of adherence to OAMs can be gleaned from the literature, the data suggest that a substantial proportion of patients struggle to adhere to these medications as prescribed, with adherence declining over time [4].

In oncohaematology, reported adherence ranges from 20% to 53% for CML patients and from 6% to 35% for ALL patients [5]. Today, apart from CML and ALL, several other haematological malignancies are increasingly treated by self-administered OAMs [6], many of which involve long and complex treatment regimens. Frequent dose adjustment due to toxicities or comorbidities, in Belgium, are associated with the non-recording of prescribed dosages and considerably limit large-scale retrospective measures of adherence in this context. This paucity in research results in an overview of factors that impact only CML and ALL patients.

In Belgium, to our knowledge, the only available data were reported by the 'ADAGIO' study [1]. Limited to CML patients, this study allowed us to recognise that, despite the gravity of their disease, as well as the fear of relapse or death, non-adherence to the oral chemotherapy medication imatinib was more frequent than physicians and their family members believed. Patient-physician interactions were innovatively investigated and identified as a contributing variable to adherence. Unfortunately, the study suffered from serious underrepresentation of patients from diverse minority populations. This is even more restricting, in that studies suggest considerable disparities and difficulties in communication between doctors and patients from different cultural and ethnic backgrounds [7–10]. Considering that medication adherence is a complex interplay of predisposing factors, patients' knowledge and beliefs and patient-physician interactions [2, 9, 10], the underrepresentation of migrants and ethnic

minorities (MEM) in adherence studies is critical. It may challenge the generalisability of the results of these cancer clinical trials to ethnic minority patients [11].

The scientific literature on health disparities mainly focuses on the use or access to healthcare services. The topic of MEM adherence to OAM is far less studied. The ethnic background has been frequently reported to be a predictor of poor adherence in other chronic diseases [12–15], but even in this field, those studies presented important limitations that prevented definite conclusions [16].

Yet, disquieting facts may raise concerns about a higher risk of poor adherence among MEM. They have been identified as one of the most vulnerable groups in health provision and have worse outcomes in cancer [17–20]. If these health disparities are largely explained by the fact that MEM are often part of the most vulnerable category of patients due to their lower educational, social and socioeconomic status, even after adjustment for socioeconomic variables, the risk of morbidity for MEM remains higher [7]. This suggests these patients are exposed to other risk factors [8, 9]. In Belgium, MEM are also confronted with barriers when using health services [21].

Disparities in health and health care not only affect the groups facing disparities, but also limit overall gains in quality of care and health for the broader population and result in unnecessary costs. Addressing health disparities is increasingly important as the population becomes more diverse [22]. The Brussels capital region has the highest concentration of population of foreign nationality in Belgium, with 34.7% foreigners on 1 January 2018 and 56.7% without Belgian nationality at birth [23]. In the context of the emergence of novel oral therapeutics in oncohaematology, it becomes urgent to identify any potential differences in the use of OAMs between our patients with or without a migrant background.

## 1.A Adherence, a concept to be considered

A number of terms, e.g. 'compliance', 'adherence', 'observance', 'persistence' or 'concordance' are currently used to define different aspects of the act of seeking medical attention, acquiring prescriptions and taking medicines appropriately. Sometimes, these are used interchangeably, but they however impose different views about the relationship between the patient and the health care provider [24].

Since the initial definition of compliance, referring to the extent to which patient behaviour coincides with clinical prescription [25], the definition of adherence has progressively evolved with an ever-increasing emphasis on patient agreement. In 2003, the World Health Organisation (WHO) defined adherence to long-term therapy as 'the extent to which a person's behaviour–taking medication, following a diet, and/or executing lifestyle changes–corresponds with agreed recommendations from a health care provider' [26]. Some continue to criticise this reductive definition as the patient seems to only have the option to agree with the subscriber's recommendations. In our opinion, this is going down the wrong path.

Adherence can be described as two complementary dimensions: adherence to medication and therapeutic adherence. The first can be defined as the degree to which the patient follows the prescribed medication. It refers to the behaviour, i.e. the measurable, visible and quantifiable part of adherence and comprises three components: 1) the initiation of treatment, referring to when the patient takes the first dose of a prescribed medication; 2) the implementation of the dosing regimen, defined as the extent to which a patient's actual dosing corresponds to the prescribed dosing regimen, from initiation until the last dose taken; and 3) discontinuation, corresponding to the end of therapy, when the next dose to be taken is omitted and no more doses are taken thereafter [24]. Therapeutic adherence rather refers to the patient's level of acceptance of the therapeutic project. This dimension recognises that the patient, as an

empowered actor, must 'adhere' to therapy and not only 'submit' to its prescription. Considering this, allows us to understand adherence not as a stable behaviour over time but rather submitted to changing environmental and psychological factors. To grasp all the stakes of care practice, we propose a mixed method able to measure both quantifiable behaviour and explore the patient's subjectivity regarding disease and treatment.

## 2. Methods/Design

### 2.A. Objectives

The objectives are: 1) to measure, in two subgroups of migrant and non-migrant patients with haematological malignancies, medication adherence and persistence regarding OAMs; 2) to identify the associated risk factors and; 3) to understand any observed differences between the two subgroups.

To meet these objectives, we developed a mixed method approach able of tackling the challenges of studying adherence in haematology and in first generation (FG) and second generation (SG) migrants of different origins.

### 2.B. Design

Conducted in the outpatient clinics of two hospitals, the MADESIO mixed-method explanatory design will combine sequentially a quantitative questionnaire-based study with an in depth qualitative approach [Fig 1]. The first four-visit questionnaire-based survey will measure adherence to OAMs and identify associated risk factors in two sub-groups of at least 60 migrant and 53 non-migrant haematological cancer patients. The second qualitative phase, using semi-structured interviews, will more deeply address the patients' subjective meanings, trying to understand any association observed in the quantitative results.

The quantitative and qualitative phases are connected on two levels. Firstly, the quantitative results will instruct the appropriate participants to be selected for the qualitative phase. Secondly, the quantitative phase will focus the results that need to be examined in more detail in the qualitative study. A six-month intermediate analysis will be performed on the quantitative results.

### 2.C. Clinical setting

The Institut Jules Bordet (IJB) and the Centre Hospitalier Universitaire Saint-Pierre (CHUStP) are two hospitals located in Brussels, the capital of Belgium, and separated by a single street.

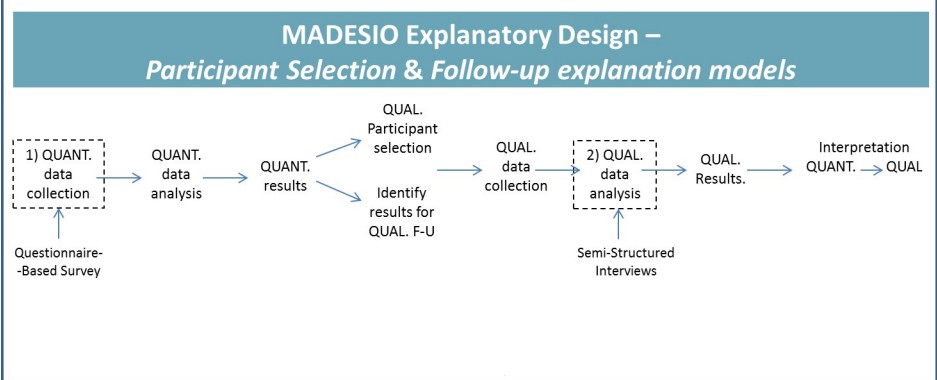

**Fig 1. MADESIO explanatory design.**

The CHUStP is a local general university hospital and the IJB is a comprehensive cancer centre involved in research and teaching. Both are part of the Iris network of Brussels public and academic hospitals whose mission is to attend every patient. To prevent any discrimination in their large population of foreign origin, they have implemented a linguistic assistance and intercultural mediation service.

Located in the centre of Brussels, their oncohaematological outpatient settings offer an ideal setting to explore adherence in the context of interethnic and intercultural medical consultations. The patients managed in oncohaematology include a high proportion of FG and SG migrants [27]. Half of them were born abroad and 40% do not have French or Dutch as their mother language. In contrast, but similar to the rest of Belgium [28], the number of foreigner doctors is limited and to a large extent from neighbouring countries and with no language barrier (the Netherlands, France, Germany) or to two traditional immigration countries for Belgium, namely Italy and Morocco (the latter also with limited language barriers since French is a widespread language).

## 2.D. Population and methods

This prospective trial is two-fold.

**2.D.1. Quantitative questionnaire-based study.** To measure adherence and identify associated risk factors, relevant data will be collected and questionnaires distributed on four successive visits, as detailed in Fig 2. For patient comfort, visits are dispersed according to the usual follow-up frequency of the patient, which depends on the haematological malignancy and disease stage. Visit 2 might be in Month 1, 2 or 3, visit 3 in Month 3, 4 or 6, and visit 4 in Month 6 or 9.

| Timepoint | Visit 1 | visit 2 | visit 3 | visit 4 |
|---|---|---|---|---|
| | Day 0 | Month 1/2/3 | Month 3/4/6 | Month 6/9 |
| **Enrolment** | | | | |
| Informed Consent | X | | | |
| **Outcome measures of adherence** | | | | |
| Tools for Adherence Behavior Screening (TABS) | X | X | X | X |
| Morisky Medication Adherence Scale (MMAS) | X | X | X | X |
| Beliefs and Behaviors Questionnaire (BBQ) | | X | | |
| Beliefs about Medicine Questionnaire (BMQ) | | X | | |
| **Additional outcome measures** | | | | |
| Human Connection Scale (HCS) | | | X | |
| Hospital Anxiety and Depression Scale (HADS) | | | X | |
| **Other variables** | | | | |
| Sociodemographic questionnaire | X | | | |
| 5-items medication questionnaire | X | | | |
| Social Desirability Scale 17 (SDS-17) | X | | | |
| Clinical and contextual informations | X | X | X | X |

**Fig 2. MADESIO questionnaire-based survey participant timeline.**

*Study population*. About 113 patients with haematological malignancies will be enrolled over 12 months, divided into two subgroups of 60 migrants and 53 non-migrants.

As the definition and operationalisation of 'migrant' will determine the categories of health inequalities and thus influence both the measurement of these inequalities and the actions taken to tackle them, defining who counts as a migrant is of crucial importance [29].

There is still no consensus on a single definition of 'migrant'. Migrants might be defined by foreign birth, by foreign citizenship or by their movement into a new country to stay temporarily (sometimes for as little as a year) or to settle for the long term [30]. If we do not consider the duration of movement, a migrant is generally accepted as any person who is moving or has moved across an international border or within a state away from his/her habitual place of residence, regardless of (1) the person's legal status; (2) whether the movement is voluntary or involuntary; (3) what the causes of movement are; or (4) what the length of stay is [31]. But, if the concept of usual residence exists and is certainly valuable for large migration statistical frameworks [32], the challenges posed by its definition makes it inoperative at the prospective clinical scale.

According to our local context and study objectives, it seems relevant to define as 'first generation (FG) migrants' the group of foreign-born persons and as 'second generation (SG) migrants' people native-born but holding either a foreign nationality or having one or both parents foreign-born. This definition, by integrating SG migrants, avoids restricting the issue to the newly arrived immigrants and allows us to address groups that have been part of the country's history for over half a century [33]. SG native-born migrants may potentially be part of a group that share minority status in Belgium due to ethnicity, place of birth, language, religion, citizenship and other cultural differences. As they may practice a different mother tongue and or different cultural norms and values from the majority culture, they may also be affected by ethnic inequalities in health.

*Sample size estimation*. The sample sizes were calculated to answer our first objective of measuring adherence to OAMs in migrants and non-migrants with haematological malignancies. In the absence of pre-existing data in migrant populations with haematological malignancies in Belgium, adherence prevalence was estimated based on mainly foreign scientific literature. The number of 60 migrant and 53 non-migrant patients were respectively calculated based on an estimated prevalence of adherence of 60% [5, 34] and 70% [3, 34, 35], considering a confidence interval of 95% with a precision of 10% and a non-response rate of 20%.

*Eligibility of participants*. Eligible patients are adult patients with a haematological malignancy, having taken at least one OAM since minimum 30 days and having a minimum of six months life expectancy. Patients should be able to read and fully understand French, Dutch, English, Arabic, Polish or Romanian. Language selection was based on Belgian and Brussels migration statistics, hospital statistics and the experience of the haematologists [36, 37].

We decided to exclude illiterate patients as they are particularly vulnerable to communication issues. Expressing oneself and reflecting on one's health condition requires some ability for abstraction which is not equally present in illiterate people. Certainly, the enrolment of these patients would have required an alternative oral qualitative approach requiring more time and human resources than what was available. By restricting the study to literate patients, we considerably reduced the need for assistance from intercultural mediators (IM). This is particularly valuable as the participation in research activities is not foreseen in the tasks of IM and their availability is extremely variable.

*Recruitment strategies*. The screening and enrolment process was conducted by a researcher who had access to consultation planning and patient medical files. All eligible patients are listed and selected in running order.

Briefly, the purpose of the study is presented during an initial phone call. If the patient agrees, a half-hour information visit is scheduled during the next regular follow-up visit. In case of a language barrier, the contact person mentioned in the patient's medical file is invited and the information visit is organised in the presence of an IM. After this information session, informed consent is obtained from the patient.

*Self-reported measures of adherence and theoretical framework.* Numerous methods for measuring adherence have been described, but no single method performs well on all criteria [38, 39]. All objective and subjective measures have advantages and disadvantages [40–47]. Whether objective or subjective, more than presenting different tools, each method appears to capture different information on medication taking [48] and should be assumed to be complementary. A combination is therefore recommended to increase the validity and reliability of the collected adherence data. Among the key elements that need to be considered when selecting self-reported measures of adherence is the fact that adherence tools must be developed based on a theoretical framework and a qualitative exploratory phase [49].

The nature, extent, and determinants of non-adherent behaviours are complex. Despite extensive research, a limited understanding of adherence phenomena and the absence of a standard theoretical framework suitable for all populations for empirically testing adherence outcomes against the determinants are common challenges for adherence research [50]. There are no theories of adherence per se, but various models and theories [51–53] exist to predict the variability that characterises behavioural adherence.

From a theoretical medical anthropology perspective, Kleinman [54] argued that health care outcomes such as adherence are directly related to the degree of cognitive disparity between the explanatory models of practitioner and patient. People's health beliefs from other cultures are often not concordant with those of western healthcare professionals, increasing beyond the language barrier the risk of misunderstanding or disagreement. Others confirm that misunderstandings are more frequent in ethnic minority patients and often result in non-adherence [8].

However, we cannot ignore the fact that there are many other theoretical models assessing other factors to explain adherence behaviours patients with migrant background. Lower socio-economic conditions, disparities in beliefs and expectations about disease or treatment and/or miscommunication between patients and physicians are a few examples of them [12]. Patients may exhibit different types of non-adherent behaviours. Unintentional non-adherence may be due to forgetfulness, or the inability to follow treatment instructions due to poor understanding or physical problems such as poor eyesight or dexterity, whereas intentional non-adherence arises when the patient rejects either the doctor's diagnosis or the doctor's recommended treatment [55, 56]. It would be biased to restrict our analysis to lower socioeconomic conditions, language discordance and cultural differences to explain any disparities regarding adherence behaviours in the migrant population.

Considering what each existing scale really measures and how they have been validated [58], we selected complementary tools assessing the factors identified as relevant domains of interest in the study of adherence behaviours, both generally and within ethnic minorities.

*Outcome measures of adherence.* Among the scales able to measure medication-taking behaviour, the **Tool for Adherence Behaviour Screening** *(TABS)* [51] and the **Morisky Medication Adherence Scale** *(MMAS-8)* [57] allow for screening both intentional and unintentional non-adherence. If the reasons behind intentional an unintentional non-adherence are distinct, the strategies for addressing them are as well. Therefore, separate measures for these two types of non-adherence are essential, especially when regression is used to determine the predictors of non-adherence.

The *MMAS-8* was designed to both measure medication-taking behaviour and to identify barriers to adherence. It has demonstrated a significant correlation with objective measures of adherence [58, 59] and it is already validated for cancer conditions in several languages. Items of the TABS are worded both positively and negatively within the same section to avoid acquiescence, affirmation or agreement bias. Furthermore, the thoughtful wording of the items allows the screening of both underutilisation and overutilisation.

These two 8-item scales have a short pass time and can easily be repeated at every visit. As medication adherence may vary and decrease over time due to a range of factors [3], this repetition is essential to evaluate treatment continuity.

Beliefs about medicines that may influence adherence must be screened, particularly when exploring adherence among culturally diverse population. On visit 2, the patient additionally completes the **Beliefs and Behaviour Questionnaire** *(BBQ)* [51] and the **Beliefs about Medicine Questionnaire** *(BMQ)* [60].

The *BBQ* appears particularly relevant for a first exploration. Not limited to any specific model for predicting non-adherence, clinically useful, simple, socially and culturally relevant, this 21 close-ended questionnaire covers the various themes of adherence in adequate depth [61]. The 18 items of the *BMQ* quantify and compare a patient's personal beliefs about the necessity of their prescribed medication and their concerns about taking it. Patients who believe their medication to be necessary and have more concerns regarding how to take it have consistently been shown to be more adherent in a range of diseases [62–65].

*Additional outcome measures*. On visit 3, the patient completes the **Human Connection Scale** *(HCS)* to measure his/her appraisal of the therapeutic alliance, i.e. the collaborative bond between the patient and the haematologist [66]. Considering that patients with a migrant background are less likely to form a therapeutic alliance with their physician [67], the patient's appraisal of interactions seems a critical domain of interest in the study of adherence in the context of interethnic or intercultural consultation.

Additionally, the patient completes the **Hospital and Anxiety Depression Scale** (HADS) [68], a simple, reliable tool to screen both anxiety and depression in people with physical health problems. Depression, anxiety, fears or anger about the illness can bring about an adverse attitude towards therapy, which can affect medication adherence [69]. Our study will invariably enrol patients with various haemopathies and prognosis, more or less further along in treatment, who experience different levels of stress or anxiety. Furthermore, migrants have specific risks and exposures to violence and stressful life experiences related to the migration process [70, 71].

*Other variables*. During the first visit, the researcher collects the full socio-demographic, economic, educational, linguistic and migratory characteristics necessary for the analysis. For patient declining to participate, we seek their consent to collect main sociodemographic data in order to assess if the respondent sample is representative of the eligible population.

Taking advantage of the presence of the IM, the researcher orally collects a **5-item medication questionnaire** exploring patient's general behaviours regarding medicines to determine whether they receive assistance in preparing or taking medicines, they use alternative or complementary medicines or have implemented changes in their lifestyle to improve their health.

At the end of the visit, the patient receives and completes an auto-administered social desirability scale. The social desirability is usually defined as '*the tendency of individuals to present themselves favourably with respect to current social norms and standard*' [72] and is considered a potential and typical bias in the measurement of self-reported adherence [73]. Considering that social desirability concerns may differ across cultures, this tendency must be captured when studying differences in self-reported health behaviours, beliefs or experience between different cultural groups. By adding the **Social Desirability Scale-17** (SDS-17) [74], we

improve our ability to assess whether observed differences may be the reflection of differences in willingness to report such behaviour or beliefs.

In parallel with every visit, clinical data are collected from patient medical records: diagnosis, disease stage, prescribed drug regimen, side effects, concomitant medications, comorbidities and performance status. Also, contextual information such as the presence of family members, language(s) used for communication, assistance of an (non)-professional interpreter, estimation of patient's level of French proficiency or the duration of the consultation, are provided by the physician.

*Strategies to enhance the quality of self-reported measures of adherence.* The quality of self-reported measures of adherence may be enhanced through efforts to use validated scales, assess the proper construct or reduce social desirability bias [75, 76]. Those were considered and implemented throughout the study design.

Firstly, we only selected questionnaires that demonstrated adequate reliability and validity in the same or similar target groups. Furthermore, compared to interview-based self-reports, we preferred auto-administrated questionnaires as they are less susceptible to information bias and interviewer effects [77].

Secondly, several efforts were made to strengthen the patient's feeling of anonymity and confidentiality. Patients are contacted and informed by the same researcher, initially trained in anthropology and familiar with interview techniques. Dressed in civilian clothes, the researcher provides the information and collects informed consent in a private consultation box, away from other patients and professionals. In parallel, questionnaires are pre-coded with the patient study number and visit date and distributed in the same Identity-coded envelope. The patient can therefore complete the questionnaires when convenient during the hospital visit and slide it into one of the numerous mailboxes available before leaving the hospital. Only the researcher can link the identity study number to the patient's name and has access to respondent answers.

Thirdly, both written and orally, the study objectives are presented without ever using the word 'adherence'. The words used further underline the willingness to understand their disease and treatment experience preventing them from feeling the need to be a 'good patient', which may lead to overinflating of adherence rates. It is key to emphasise that there is no judgment about taking or not taking their treatment as prescribed and to indicate that our interest is about their feelings and experience. With the same objective, each adherence scale starts with a statement normalising non-adherence, recognising the challenges of taking regular medications.

Finally, to avoid any unwilling changes in haematologist attitudes or any increased motivation related to the Hawthorne effect, patient enrolment and data processing are wholly independent of them. For the same reason, participating haematologists ignore the content of the questionnaires.

*Translation and validation of selected instruments.* Although using an existing validated questionnaire will save time and resources, it may not be readily available in the language required for the targeted respondents [78]. Among the selected questionnaires, only the HADS was already validated in six translated versions. For others, we established a rigorous, appropriate and user-friendly valid translation method. According to several recommendations, we combined different translation techniques and applied the steps inspired by WHO guidelines and described in Fig 3 [78–82].

1)The original English source questionnaire was translated by a linguistic and culturally competent professional translator. 2) Another independent translator converted back the translated version to English and 3) the researcher ranked each item in terms of comparability of language and interpretability. All discrepancies, misunderstandings or unclear wordings

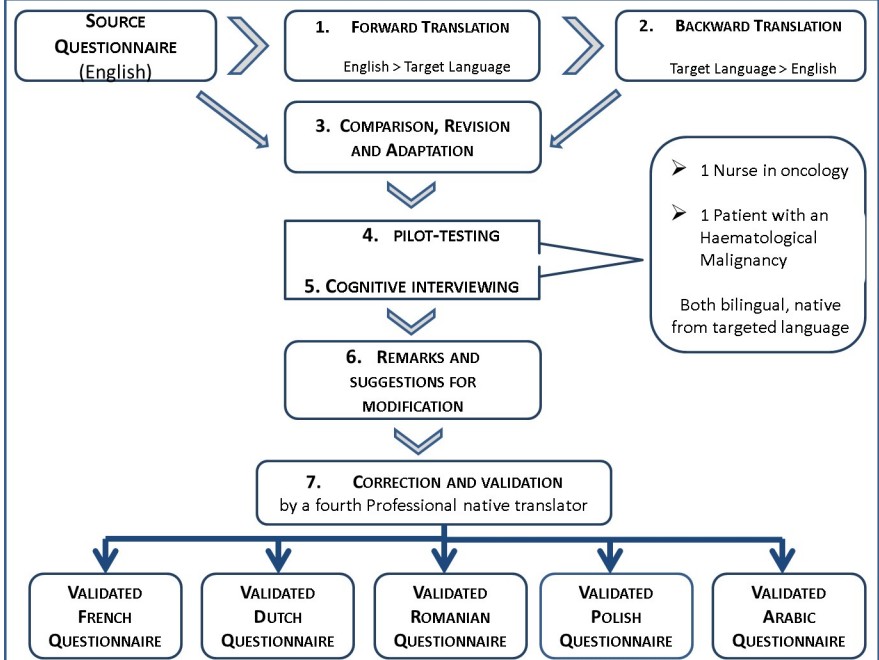

**Fig 3. Translation and validation process of selected instruments.**

were discussed with a third independent translator and retranslated if needed. 4) The pre-final translated version was pilot-tested with one bilingual patient with a haematological malignancy and one bilingual nurse in oncology. 5) A cognitive debriefing allowed to check if the translated items retained the same meaning as the original, and to assess the clarity and appropriateness of wording. 6) All suggestions for linguistic and cultural adaptation were discussed with the researcher, the two bilingual respondents and the third professional translator until a consensus was reached (7).

In some cases, the return of the instrument to the authors was necessary to clarify the original meaning of a statement; in other cases, translated versions were adapted to treatment or language specificities.

The MMAS-8 was originally built to assess medication-taking behaviours in chronic diseases and includes the following question: 'Did you take your medication yesterday?'. Haematological cancer patients might face various intricate medication regimens. Some take an OAM only 21 days on 30, so that if the study visit is scheduled during the 7 interruption days, the question is not relevant to measuring adherence. As patients always came in for consultation on the first day of their new 21-day cycle, the question was changed to 'Did you take your medication today?'

The terms 'cancer' and 'oral anticancer medication' were respectively changed to 'haematological disease' and the patient's specific drug name. This allowed us to bypass the dilemma existing in some languages or cultures regarding the use of the word 'cancer' while respecting the patient's perception, who may not define their malignancy as a cancer but as a disease.

The interpretability of the BBQ item 'I put up with my medical problems before taking any action' was unobvious. The confusing interpretation of this item was not revealed by the forward and back-translation process. But, during the cognitive interviews, the meanings given by the respondents varied greatly. Some understood it as making an effort to not take a medicine for unpleasant symptoms, and by others as making an effort to not seek medical care

when they have a medical problem. Others wanted to ensure that all their cancer-related problems were under control before starting any sport activity. After clarifying the meaning with the developers of the questionnaire, we changed the original item to: 'When I experience medical problems, I delay seeking medical treatment or help from doctors'. The authors of BBQ think that more adherent patients are more likely to seek medical care (as opposed to self-treatment) in the presence of symptoms, and that worry about health in general is positively correlated with action.

Finally, the BBQ item 'It is physically difficult to handle some of my medications' also required clarification from the developers. After backward translation, 'to handle' became 'to take' in Polish and 'to use' in Dutch, with the related risk of some interpreting this as difficulty with manipulating the medication or others as difficulty with swallowing. The BBQ authors confirmed the original meaning of the item referring to the 'difficulty with handling my medication due to dexterity issues'.

Now that the translated versions are no longer confusing, and the meaning has been preserved, the final version will be implemented for all patients enrolled in the study.

*Statistical analysis.* Data will be recorded using the Survey Processing System 7.4© and will be analysed using STATA version 15. Descriptive statistics will be performed. For each subgroup, the mean and standard deviation (SD) or the median and interquartile range, depending on the distribution, will be calculated for each score. For categorical variables, Pearson's chi-squared test will be used; for continuous variables, Student's t-test will be used. P-values <0.05 will be considered significant.

**2.D.2. Qualitative semi-structured interviews.** Quantitative research methods have significant limitations in capturing the complexity of human behaviour and experience, tending to ignore the social and discursive contexts in which individual and collective understandings of illness experience emerge [83]. Understanding the meaning that patients give to their illness and treatment experience may be crucial to explain the statistical associations between variables observed in the quantitative study.

At this stage, we can only hypothesise that adherence rates or associated risk factors may be different between the subgroups of migrants and non-migrants. In any case, the qualitative study will help us to understand why these results differ or not.

*McGill Illness Narrative Interview.* The McGill Illness Narrative interview (MINI) was designed to elicit the individual meaning of illness, the modes of reasoning, the historical sequences and the sociocultural contexts of illness experience [84]. Addressing the discursive contexts in which individual and collective understandings of illness experience emerge may contribute to understanding any potential association between personal or cultural beliefs and adherence behaviours.

The semi-structured framework of the MINI allows for a wide range of interpretative strategies, from critical and interpretative anthropology [85] to literary theory [84, 86, 87] via grounded theory [88]. At this stage and given the lack of previous existing data in similar population or context, the fact that theory can inductively emerge from collected data is an asset. If Kleinman [54] argues that medication adherence is directly related the degree of cognitive disparity between the explanatory models of practitioner and patient, Weiner [89] reminds us that patients facing a serious illness such as haematological malignancy do not always offer causal attributions for their illness. Lay accounts of illness experience not always form logical and coherent schemas organised around causal attribution. Therefore, Groleau in '*Déterminants culturels et l'approche écologique*' underlines that restrict explanatory models to patients' causal attributions of their disease may only reveal a small portion of the many representations that come into play with regards to illness and health-related behaviour. Thanks to the MINI, we will be able to elicit three distinct types of reasoning about symptoms or illness experience:

1) a basic narrative account structured by contiguity, 2) a prototype narrative, focusing on previous experiences with similar conditions, and 3) an explanatory model narrative organised in terms of explicit knowledge of causes, mechanisms or other cultural models of process.

Unstructured elements of the MINI will provide useful narratives to study individual subjective meanings, including the patient's level of acceptance of the therapeutic project. Contrarily, the structured dimension allows us to compare the sociocultural contexts of illness experience between groups of patients from different migrant backgrounds.

*Sample size and selection.* According to a sequential mixed-method explanatory design, an intermediate analysis of quantitative results is needed to purposively select appropriate participants for sample stratification. At this stage, we can consider selecting outliers from the regression analysis or decide to learn more about the differences between people who score either high or low on adherence variables.

This purposive sampling will be followed by concurrent data generation and data analysis. Through various stages of coding, undertaken in conjunction with constant comparative analysis, we will employ theoretical sampling until theoretical saturation is reached.

Eligible patients will be identified, informed, and enrolled with the same process as for the quantitative study. The researcher will conduct the interviews for 45–90 minutes in a private room, with the patient alone or in the presence of the IM when needed.

*Analysis.* The interviews will be audio taped and transcribed verbatim. The narratives will be analysed according to their form or structure or their content at a collective level, exploring any recurrent themes or structures among narrators.

At this stage and in the absence of previous existing data, we cannot assume any common patterns of meaning in specific groups of FG or SG migrants. But, an individual's understanding of illness may be viewed as a co-construction of the meaning and reflects at the same time 1) the previously acquired and organized interpretations of illness, 2) new reflections of illness experience and 3) the unfolding relationship between the interviewer and the respondent [6]. Therefore, if the MINI does not distinguish between personal and cultural meanings linked to disease and treatment, the distinction between this level of knowledge can be determined by adding explicit questions or by comparing interviews across groups of individuals from specific cultural backgrounds or social positions.

**2.D.3. Ethics approval and consent to participate.** The MADESIO protocol has been submitted and approved by the Institut Jules Bordet Academic Trials Promoting Team (Comité des Projets Cliniques–CPC) on the 24.04.2019 and both by the Ethics Committee (EC) of the Institut Jules Bordet (accreditation number OM011) as central EC on the 11.07.2019 and the EC of the Centre Hospitalier Universitaire Saint-Pierre (accreditation number OM011) as local EC on the 03.09.2019, with the reference number 2030. A Privacy Impact Assessment (PIA) has been issued in accordance with GDPR regulations and validated with a minor impact by the two Data Protection Officers of both hospitals. Written informed consent will be sought from patients before starting to participate. All data will be securely stored and password protected, accessible only to the researcher. Only pre-coded data will be transferred to the research team and only disaggregated data will be used in publications resulting from this study. Protocol details are recorded and amendments will be reported to ClinicalTrials.gov. The research team will follow the ethical standards of the Declaration of Helsinki.

## 3. Discussion

The MADESIO Protocol was designed to tackle the challenges of studying adherence in patients with various haematological malignancies and with diverse language and migrant backgrounds.

Based on broad theoretical approach, our study will suit to migrant populations by exteding the explored adherence predictive factors to contextual variables, miscommunication issues and factors shaped by culture.

Methodologically, we implemented diverse strategies to enhance the quality of data collected.

We combined validated scales to strengthen the reliability and validity of self-reported measures. Based on a rigorous translation and validation process we ensured robust data comparability for our sample. Throughout the selection, the distribution and the collection, we worked to minimize the social desirability bias and strengthen the patients' feeling of anonymity and confidentiality. In parallel, the questionnaire distribution process was designed to not increase the patient's burden, and therefore enhance their participation and reduce the non-response rate. Strategically, we also favored auto-administrated questionnaires with closed-ended questions in order to reduce the necessary human ressources for data collection of translation assistance. Finally, with our qualitative approach we will overcome the limits of descriptive analysis and provide a comprehensive understanding of adherence behaviours.

Definitely, a number of limitations can be identified. First of all, our study is exploratory with a small sample size. The comparison of observed adherence rates between the migrant and non-migrant subgroups will not be sufficient for statistical significance. A larger sample would increase the representativeness of the results and limit any centre effect. Furthermore, the validation of the translation process, even rigorous, is limited by the restricted available resources. The bilingual individuals involved in the process represent a separate population that cannot be automatically generalised to the monolingual target population [85]. The translation would benefit from being validated with a pilot phase on larger monolingual patient samples.

The upcoming MADESIO mixed-method study will provide a first overview of how non-migrants and migrants with haematological malignancies, in our two Brussels Hospitals, adhere to their OAMs. This comprehensive approach will offer a solid starting point to see beyond the identification of language and ethnicity as risk factors and better understand whether and why FG and SG migrants constitute a risky population regarding adherence to OAMs. The knowledge generated may be extremely valuable for healthcare services in order to simplify the development of patient-tailored complex interventions and to guide providers in choosing the strategies to address medication adherence of migrants or diverse minority populations.

Singularly, our protocol may highlight the feasibility for healthcare services to fight against cancer inequities by routinely detecting poor adherence in diverse minority populations. By disseminating this protocol in the peer-reviewed literature and in national and European scientific forums, we could convince haematologists and young investigators to be part of the next generation of haemato-oncologists who put ethnicity and culture in the scope of cancer research. A collaborative action from the community of clinicians and researchers will be highly beneficial for the sharing of cross-culturally adapted instruments and protocols.

## Acknowledgments

We warmly thank all patients who participated in the study. More particularly, we thank the bilingual patients and nurses who participated in the preliminary pilot testing. We thank all the haematologists who supported the research and agreed to answer to the clinical and contextual questions throughout the study visits: Prof N. Meuleman, Prof P. Lewalle, Dr A. Salaroli, Dr C. Spilleboudt, Dr M. Maerevoet, Dr M. Vercruyssen, Dr S. Wittnebel, Dr F. Massaro, Dr F. Andreozzi.

We thank also Mr. Ibrahima Diallo for the developpement and programmation of the Survey Processing System allowing encoding and registration of the quantitative data collected. We thank MMAS Research LLC. for providing the 3rd July 2018, upon payment of licence fees, the right to use the MMAS-8 form. We thank Prof J. George from Monash University who gave us the authorisation to use the TABS and BBQ free of charge (authorisation obtained on 15 May 2018). We thank Prof R. Horne from University College London, who gave us the permission to use the BMQ without charge (authorisation obtained on 28 August 2019). We thank Prof J. W. Mack from the Harvard University and Prof H.G. Prigerson from Weill Cornell Medicine Centre for Research on End-of-Life Care who gave their agreement to use the HCS without charge (authorisation obtained on 3 July 2018). We also thank the GL Assessment team for their support in obtaining the licence to use the Hospital Anxiety and Depression Scale against payment of license fees (permission obtained on 26 November 2019, ref. PRF0002647). We thank Prof Stoeber who allowed us to use the SD-17 free of charge as of 12 February 2019. Finally, we would like to thank Prof D. Groleau and Dr L.J. Kirmayer from McGill University who authorised us to use the McGill semi-structured Interview on 10 July 2018.

## Author Contributions

**Conceptualization:** Sandra Michiels, Sandra Tricas-Sauras, Marie Dauvrin, Dominique Bron, Fati Kirakoya-Samadoulougou.

**Funding acquisition:** Sandra Michiels, Dominique Bron.

**Methodology:** Sandra Michiels, Sandra Tricas-Sauras, Marie Dauvrin, Fati Kirakoya-Samadoulougou.

**Project administration:** Sandra Michiels.

**Resources:** Dominique Bron.

**Supervision:** Sandra Tricas-Sauras, Marie Dauvrin, Fati Kirakoya-Samadoulougou.

**Writing – original draft:** Sandra Michiels.

**Writing – review & editing:** Sandra Michiels, Sandra Tricas-Sauras, Marie Dauvrin, Dominique Bron, Fati Kirakoya-Samadoulougou.

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
